# Long Non-Coding RNA CRNDE Is Involved in Resistance to EGFR Tyrosine Kinase Inhibitor in EGFR-Mutant Lung Cancer via eIF4A3/MUC1/EGFR Signaling

**DOI:** 10.3390/ijms22084005

**Published:** 2021-04-13

**Authors:** Satoshi Takahashi, Rintaro Noro, Masahiro Seike, Chao Zeng, Masaru Matsumoto, Akiko Yoshikawa, Shinji Nakamichi, Teppei Sugano, Mariko Hirao, Kuniko Matsuda, Michiaki Hamada, Akihiko Gemma

**Affiliations:** 1Division of Pulmonary Medicine and Oncology, Graduate School of Medicine, Nippon Medical School, Bunkyo-ku, Tokyo 113-8602, Japan; satoshi506@nms.ac.jp (S.T.); r-noro@nms.ac.jp (R.N.); s7062@nms.ac.jp (M.M.); s9056@nms.ac.jp (A.Y.); snakamichi@nms.ac.jp (S.N.); tetus629@nms.ac.jp (T.S.); m-hirao@nms.ac.jp (M.H.); kuniko-m@nms.ac.jp (K.M.); agemma@nms.ac.jp (A.G.); 2Department of Electrical Engineering and Bioscience, Faculty of Science and Engineering, Waseda University, Shinjuku-ku, Tokyo 169-8555, Japan; zengxchao@gmail.com (C.Z.); mhamada@waseda.jp (M.H.); 3AIST-Waseda University Computational Bio Big-Data Open Innovation Laboratory (CBBD-OIL), Shinjuku-ku, Tokyo 169-8555, Japan

**Keywords:** EGFR-TKI, drug resistance, CRNDE, eIF4A3

## Abstract

(1) Background: Acquired resistance to epidermal growth factor receptor-tyrosine kinase inhibitors (EGFR-TKIs) is an intractable problem for many clinical oncologists. The mechanisms of resistance to EGFR-TKIs are complex. Long non-coding RNAs (lncRNAs) may play an important role in cancer development and metastasis. However, the biological process between lncRNAs and drug resistance to EGFR-mutated lung cancer remains largely unknown. (2) Methods: Osimertinib- and afatinib-resistant EGFR-mutated lung cancer cells were established using a stepwise method. A microarray analysis of non-coding and coding RNAs was performed using parental and resistant EGFR-mutant non-small cell lung cancer (NSCLC) cells and evaluated by bioinformatics analysis through medical-industrial collaboration. (3) Results: Colorectal neoplasia differentially expressed (CRNDE) and DiGeorge syndrome critical region gene 5 (DGCR5) lncRNAs were highly expressed in EGFR-TKI-resistant cells by microarray analysis. RNA-protein binding analysis revealed eukaryotic translation initiation factor 4A3 (eIF4A3) bound in an overlapping manner to CRNDE and DGCR5. The CRNDE downregulates the expression of eIF4A3, mucin 1 (MUC1), and phospho-EGFR. Inhibition of CRNDE activated the eIF4A3/MUC1/EGFR signaling pathway and apoptotic activity, and restored sensitivity to EGFR-TKIs. (4) Conclusions: The results showed that CRNDE is associated with the development of resistance to EGFR-TKIs. CRNDE may be a novel therapeutic target to conquer EGFR-mutant NSCLC.

## 1. Introduction

Epidermal growth factor receptor-tyrosine kinase inhibitors (EGFR-TKIs) have shown marked efficacy in non-small cell lung cancer (NSCLC) patients with EGFR mutations [1,2,3,4,5,6]. Osimertinib (a third-generation EGFR-TKI) showed significant efficacy in NSCLC patients with and without the T790M mutation in exon 20, which was previously considered a second-hit mutation [7,8]. Acquired mechanisms of resistance after treatment with osimertinib, including C797S mutation, activation of bypass pathway such as MET amplification, BRAF mutation, and transformation of small cell lung cancer have been previously reported [9]. However, other mechanisms of resistance to EGFR-TKIs remain unclear.

In addition to mutations and aberrant expression of protein-encoding genes, non-coding RNAs (including microRNAs) appear to play a major role in cancer. We have previously reported that the miR-21, miR-134/487b/655, and miR-200 families can be used as therapeutic targets associated with EGFR-TKI treatment [10,11,12]. Long non-coding RNAs (lncRNAs), defined as transcripts longer than 200 nucleotides, are unable to encode proteins. LncRNAs play an important role in the development of tumors in a variety of human cancers, including lung cancer. Recently, lncRNAs have been shown to regulate gene expression and participate in multiple biological processes, such as cell proliferation, migration, metastasis, apoptosis, and drug resistance [13,14]. Moreover, they exhibit tumor suppression and carcinogenic functions and have been identified as a promising novel biomarker and therapeutic target for lung cancer. However, there is only limited knowledge of their roles in biological processes and drug resistance in lung cancers with EGFR mutations.

In this study, we used comprehensive lncRNA expression profiles and medical–industrial bioinformatics techniques to identify lncRNAs associated with the mechanism of acquired resistance to EGFR-TKIs in EGFR-mutant NSCLC cells.

## 2. Results

### 2.1. Effects of Afatinib and Osimertinib on Lung Adenocarcinoma Cells and Establishment of Resistant Cell Lines

We evaluated the anti-tumor properties of afatinib and osimertinib in four EGFR-mutant lung adenocarcinoma cell lines (i.e., PC-9, HCC827, H1975, and H1650) by the 3-(4,5-dimethylthiazol-2-yl)-5-(3-carboxymethoxyphenyl)-2-(4-sulfophenyl)-2H-tetrazolium, inner salt (MTS) assay. Whenever PC-9 and HCC827 cells were sensitive to afatinib and osimertinib, (IC_50_: PC-9 afatinib: <0.001 μM, osimertinib: 0.004 ± 0.0008 μM and HCC827 afatinib: <0.001 μM, osimertinib: <0.001 μM), H1975 and H1650 cells were resistant to these agents (IC_50_: H1975 afatinib: 3.3 ± 1.9 μM, osimertinib: 0.136 ± 0.043 μM and H1650 afatinib: 2.3 ± 0.33 μM, osimertinib: 2.1 ± 0.27 μM) (Table 1).

We established afatinib-resistant (AR) or osimertinib-resistant (OR) cell lines from the parental lines PC-9 and HCC827 through a stepwise method [12,15]. After six months, we established four drug-resistant cell lines: PC-9-AR, PC-9-OR, HCC827-AR, and HCC827-OR. We confirmed the stability of the cell lines that became resistant for at least 12 months without the addition of any reagents. We also compared the dose-dependence of the parental cell lines and the established resistant cell lines to each drug. The IC_50_ values of the resistant cell lines were almost 1000 times higher than those of the parental cell lines (Table 1). These resistant cell lines did not have secondary mutations, such as T790M, C797S, and MET amplification [12].

We examined the protein expression levels of EGFR signaling pathway molecules in parental cells and EGFR-TKIs-resistant cells by Western blotting. We observed decreased protein levels of phospho-EGFR (p-EGFR) in AR and OR cells and increased protein levels of p-AKT and p-extracellular signal-regulated kinase (p-ERK) in the three established resistant cells, except for HCC827-AR (Figure 1).

### 2.2. Overexpression of Colorectal Neoplasia Differentially Expressed (CRNDE) in EGFR-TKI-Resistant Cell Lines

We next performed non-coding and coding RNA microarray analyses to evaluate the post-transcriptional regulation associated with drug resistance using the parental and EGFR-TKI-resistant cells. In total, we analyzed the expression of 27,246 messenger RNAs (mRNAs) and 25,945 lncRNAs (Appendix A). We also normalized differences in the expression of mRNAs and lncRNAs (Appendix A).

We mapped mRNAs, lncRNAs, and other RNAs based on available bioinformatics tools and databases that are useful in the search for non-coding RNAs and analysis of their structure, conservation, interactions, co-expression, and localization [16]. Candidate lncRNAs and mRNAs were determined by the difference in expression according to a cutoff value. Of note, they exhibited expressions more than twice as high as those of resistant and parental cells (Figure 2A). Figure 2B shows the top 57 such lncRNAs.

RNA-protein binding data showed potentially mutual relationships of a certain RNA and protein. Of the 57 lncRNAs, CRNDE and DiGeorge syndrome critical region gene 5 (DGCR5) would bind in an overlapping manner to five proteins: eukaryotic translation initiation factor 4A3 (eIF4A3), fragile X mental retardation protein (FMRP), FUS RNA binding protein (FUS), lin-28 homolog A (LIN28A), and UPF1 RNA helicase and ATPase (UPF1) (Table 2). Accordingly, we examined the expression of CRNDE and DGCR5 in resistant and parental cells. These lncRNAs were more highly expressed in intrinsically resistant cells (e.g., H1975: afatinib resistant cells and H1650: afatinib/osimertinib resistant cells) than in sensitive cells (e.g., PC-9 and HCC827 cells) (Figure 2B). The expression of CRNDE and DGCR5 was also evaluated in established resistant cell lines by quantitative reverse transcription-polymerase chain reaction (qRT-PCR) analysis. Significant increased expression of CRNDE was observed in PC-9 vs. PC-9-AR(CRNDE: PC-9 vs. PC-9-AR, *p* < 0.01). Other than in this set, the expression of CRNDE or DGCR5 tended to be upregulated, although not significantly (CRNDE: PC-9 vs. PC-9-OR, *p* = 0.07; HCC827 vs. HCC827-AR and HCC827-OR, *p* = 0.10 and *p* = 0.11, respectively; DCGR5: PC-9 vs. PC-9-AR and PC-9-OR, *p* = 0.31 and *p* = 0.23, respectively; HCC827 vs. HCC827-AR and HCC827-OR, *p* = 0.59 and *p* = 0.73, respectively) (Figure 2C,D).

Next, we explored a specific protein binding to these lncRNAs. These lncRNAs were presumed to bind directly to eIF4A3, which was the most abundant protein in EGFR-TKI-sensitive cells (Figure 3A). Therefore, we assessed the expression of eIF4A3 in EGFR-TKI-resistant cells by Western blotting. The levels of eIF4A3 were decreased in EGFR-TKI-resistant cells, compared with the parental cells (Figure 3B). We subsequently assessed the protein expression of eIF4A3 downstream signaling molecules in parental and EGFR-TKI-resistant cells. The analysis showed that the levels of mechanistic target of rapamycin kinase (mTOR), programmed cell death 4 (PDCD4), and p-p70-S6 were increased in EGFR-TKI-resistant cells. In contrast, those of mucin 1-C (MUC1-C) and MUC1 were decreased in the resistant cells (Figure 3C).

### 2.3. CRNDE Inhibition Induced Apoptotic Activity and Overcame the Resistance to EGFR-TKIs

We next assessed the effects of these lncRNAs on the restoration of eIF4A3 expression, apoptotic activity, and sensitivity to EGFR-TKIs using specific small-interfering RNAs (siRNAs). Figure 4A shows that transfection with CRNDE siRNA inhibited the expression of CRNDE in HCC827-OR cells by 90%. Knockdown of CRNDE was achieved by the inhibition of eIF4A3 and eIF4A3 downstream signaling molecules, such as MUC1-C protein (Figure 4B). We also observed an increase in p-EGFR expression upon inhibition of CRNDE (Figure 4B). In contrast, the protein levels of the AKT/mTOR pathway were unchanged, suggesting that CRNDE independently affected the expression levels of eIF4A3 protein (Figure 4B). In addition, protein levels of eIF4A3 and the molecules of eIF4A3 downstream signaling and AKT/mTOR pathway were changeless by transfection with DGCR5 siRNA (data not shown).

We also assessed the apoptotic activity induced by CRNDE inhibition combined with osimertinib treatment. After silencing CRNDE, the protein levels of cleaved poly (ADP-ribose) polymerase (PARP) and cleaved caspase 3 (CASP3) increased (Figure 4C). Annexin V analysis demonstrated that inhibition of CRNDE also increased the number of apoptotic cells at various doses of osimertinib, compared with a control siRNA (Figure 4D).

Finally, we evaluated whether inhibition of CRNDE restores sensitivity to EGFR-TKIs in EGFR-mutant NSCLC cells. We measured the response to EGFR-TKIs in HCC827-AR and HCC827-OR cells after treatment with CRNDE siRNA. Knockdown of CRNDE promoted sensitivity to EGFR-TKIs in HCC827-OR and HCC827-AR (Figure 4E). Therefore, inhibition of CRNDE may activate the eIF4A3/MUC1/EGFR signaling pathway and restore sensitivity to EGFR-TKIs (Figure 5).

## 3. Discussion

Using non-coding and coding RNA microarray analysis, we have identified two lncRNAs (CRNDE and DGCR5) associated with resistance to EGFR-TKIs in EGFR-mutant NSCLC cells. We successfully identified the target protein of CRNDE and DGCR5, namely eIF4A3, by a bioinformatic technique. It has been reported that lncRNAs function at three levels: transcriptional, post-transcriptional, and epigenetic regulatory [14]. However, the mechanisms by which lncRNAs directly or indirectly promote carcinogenesis, metastasis, and drug resistance have not been elucidated. The interactions of lncRNAs with their coding genes and proteins remain unclear. In this study, we examined these interactions based on the mechanisms associated with resistance to EGFR-TKIs. Direct target discovery of lncRNAs using bioinformatics techniques through a medical-industrial collaboration is innovative and creative.

RNA-protein binding data showed that eIF4A3 bound to CRNDE and DGCR5 lncRNAs. Previous studies reported that DGCR5 plays an important role in the development of lung cancer [17,18]. In the present study, despite DGCR5 overexpression, silencing of DGCR5 did not alter the eIF4A3/MUC1 signal or p-EGFR, and did not significantly reduce the IC_50_ value of osimertinib with or without transfection of DGCR5 siRNA (data not shown). Thus, we did not select DGCR5 as a candidate lncRNA.

CRNDE is an oncogenic lncRNA located at an atypical locus—hCG_1815491 on chromosome 16—and was activated early in colorectal cancer [19]. It has been shown to indirectly affect oncogenic proteins via some microRNAs in lung cancers [20,21]. The expression of CRNDE in EGFR-TKI-resistant cells was higher than that measured in parental EGFR-TKI-sensitive cells. Decreased expression of eIF4A3, regulated by PDCD4 activation, was observed in these EGFR-TKI-resistant cells. Furthermore, the levels of MUC1 and p-EGFR were diminished in these resistant cells. CRNDE downregulates eIF4A to express MUC1 and p-EGFR regardless of the upstream status of eIF4A. 

EIF4A3 is an RNA-binding protein and a core component of the exon junction complex [22], which plays important roles in mRNA splicing, transport, translation, and RNA decay. In particular, it is considered that eIF4A3 is essential for nonsense-mediated decay [23], because its knockdown induced a defect in nonsense-mediated decay [24]. A recent study suggested that overexpression of eIF4A3 was related to poor prognosis, while its dysfunction promoted tumor cell migration, invasion, and drug resistance [25]. Some studies reported that eIF4A3 bound to lncRNAs and was involved in the development of cancer [26,27].

Overexpression of MUC1-C in cancer cells was also repressed by inhibiting the eIF4A RNA helicase activity. MUC1-C is a large and heavily glycosylated transmembrane protein that lubricates and protects the cell surface and increases cell signaling via EGFR [28,29]. CRNDE significantly promotes cell proliferation by mediating multiple signaling pathways and various target genes, including the PI3K/AKT signaling pathway [19]. Furthermore, knockdown of CRNDE recovered sensitivity to EGFR-TKI and exerted a synergistic apoptotic effect. Inhibition of CRNDE induced the activation of the eIF4A/MUC1/EGFR pathway and may activate EGFR signaling (Figure 5). Hence, CRNDE may be a promising therapeutic target in patients with EGFR-TKI-resistant EGFR-mutant NSCLC.

In summary, CRNDE induced resistance to EGFR-TKI via downregulation of eIF4A-MUC1 signal transduction and may be a molecular target for overcoming resistance to EGFR-TKI. Further studies will be performed to confirm the expression of CRNDE in post-progression clinical samples after treatment with EGFR-TKI. Inhibition of CRNDE could be a promising novel therapeutic strategy for patients with EGFR-mutant NSCLC to conquer the resistance to EGFR-TKI.

## 4. Materials and Methods

### 4.1. Cell Cultures

Four human lung adenocarcinoma cell lines were used in this study. PC-9 with an exon 19 in-frame deletion was provided from Immuno-Biological Laboratories (Gunma, Japan). NCI-HCC827 (HCC827) with a deletion in exon 19, NCI-H1975 (L858R/T790M), and H1650 with a deletion in exon 19 and PTEN null were purchased from the American Type Culture Collection (Manassas, VA, USA). These cell lines were cultured in RPMI1640 (FUJIFILM Wako Pure Chemical Industries, Osaka, Japan), including 10% fetal bovine serum (BioWest, Nuaillé, France) and 1% penicillin and streptomycin (FUJIFILM Wako Pure Chemical Industries), at 37 °C in a 5% CO2 incubator as previously described [12]. All cells were constantly examined for the absence of mycoplasma, and each experiment was performed independently three times for each condition.

### 4.2. Drugs and Cell Viability Assay

Afatinib and osimertinib were obtained from Selleck Chemicals (Houston, TX, USA). We used the MTS assay to evaluate the sensitivity of the human lung adenocarcinoma cell lines to afatinib and osimertinib as previously described [12]. For the MTS assay, 5000 cells per well were seeded in 96-well tissue culture plates and incubated for 24 h. Subsequently, the cells were treated with various concentrations of EGFR-TKIs or vehicle (dimethyl sulfoxide) at 37 °C for 72 h. Viability experiments were performed using the Cell Counting Kit 8 (Promega Corporation, Madison, WI, USA) and a microplate reader (Infinite M200 PRO; Tecan Group Ltd., Männedorf, Switzerland) in accordance with the manufacturer’s instructions. The IC_50_ value was defined as the concentration of drug required for 50% inhibition of growth. The corrected absorbance of each sample was calculated by comparing with that of the untreated control. Each experiment was carried out thrice.

### 4.3. Western Blotting Analysis

Protein extraction, two-dimensional polyacrylamide gel electrophoresis, and transfer to nitrocellulose membrane were carried out as previously described [11,12,15]. The membrane was incubated with the following antibodies: cleaved CASP3, cleaved PARP, PARP, p-EGFR, EGFR, p-AKT, AKT, p-ERK, ERK, p-STAT3, STAT3, eIF4A3, p-4E-BP1, mTOR, p-P70-S6, MUC1, and MUC1-C, purchased from Cell Signaling Technology (Danvers, MA, USA). The antibody to glyceraldehyde-3-phosphate dehydrogenase was purchased from Santa Cruz Biotechnology (Santa Cruz, CA, USA).

### 4.4. RNA Extraction

Total RNA from cells was extracted by TRIzol Reagent (Thermo Fisher Scientific, Waltham, MA, USA) as previously described [11,30] or ISOGEN (NIPPON GENE, Tokyo, Japan).

### 4.5. Microarray Analysis

Gene expression microarray analysis was performed using a SurePrint G3 Human Gene Expression 8 × 60 K v3 (Takara Bio, Shiga, Japan). RNA from cells was extracted using the NucleoSpin RNA Kit (MACHEREY-NAGEL GmbH & Co. KG, Düren, Germany), following the manufacturer’s recommendations. Cyanine-3 (Cy3) labeled cRNA was prepared from 0.1 ug total RNA using the Low Input Quick Amp Labeling Kit (Agilent) in accordance with the manufacturer’s instructions, followed by RNeasy column purification (QIAGEN, Valencia, CA). Dye incorporation and cRNA yield were checked with a NanoDrop ND-2000 spectrophotometer. A 0.6 ug quantity of Cy3-labeled cRNA was fragmented at 60 °C for 30 min in a reaction volume of 25 ul containing 1× Agilent fragmentation buffer and 2× Agilent blocking agent, following the manufacturer’s instructions. On completion of the fragmentation reaction, 25 ul of 2× Agilent hybridization buffer was added to the fragmentation mixture and hybridized to SurePrint G3 Human GE 8 × 60 K Microarray Ver 2.0 (Agilent) for 17 h at 65 °C in a rotating Agilent hybridization oven. After hybridization, microarrays were washed for 1 min at room temperature with GE Wash Buffer 1 (Agilent) and 1 min with 37 °C GE Wash buffer 2 (Agilent). Slides were scanned immediately after washing on the Agilent SureScan Microarray Scanner (G2600D), using one color scan setting for 8 × 60 K array slides (scan area 61 × 21.6 mm, scan resolution 3 um, dye channel set to Green PMT at 100%). The scanned images were analyzed with Feature Extraction Software 11.5.1.1 (Agilent) using default parameters to obtain background subtracted and spatially detrended processed signal intensities. Value definition was normalized signal intensity.

### 4.6. Real-Time qRT-PCR

The cDNA of CRNDE, DGCR5, and eIF4A3 was used for qRT-PCR analysis using the THUNDERBIRD SYBR qPCR/RT Set III (TOYOBO, Osaka, Japan) in accordance with the manufacturer’s instructions. The expressions of these genes were examined through the TaqMan Gene Expression Assay (Thermo Fisher Scientific) [31,32]. Gene expression levels were quantified using the 2−ΔΔCt method.

### 4.7. siRNA Transfection

The siRNA targeting CRNDE and negative control were purchased from Thermo Fisher Scientific. All siRNAs were treated with Lipofectamine RNAiMAX (Thermo Fisher Scientific) transfection reagent 24 h after seeding, according to the instructions provided by the manufacturer. We transfected the siRNA complexes into cells at a final concentration of 50 nM. Six hours after the addition of the siRNA complexes, cells were seeded in RPMI1640 containing 10% fetal bovine serum, 1% penicillin and streptomycin, and incubated at 37 °C for 48 h.

### 4.8. Statistical Analysis

Differences in categorical outcomes were assessed by the chi-squared test. The statistical significance of differences was decided with the standard Student’s t-test. A *p*-value of <0.05 was considered statistically significant. Analyses were carried out by the statistical software JMP 9 (SAS Institute, Cary, NC, USA).

### 4.9. Bioinformatics

The transcriptome (consisting of mRNAs and lncRNAs) was used as a reference for mapping. LncRNA sequencing data were extracted from LNCipedia [33] (https://lncipedia.org/downloads/lncipedia_5_0.fasta, accessed on 16 May 2018), including Refseq, Ensembl, Gencode, Broad Institute (Human Body Map lincRNAs), NONCODE, and FANTOM CAT, to establish a lncRNA database. The mRNA sequencing data were extracted from Gencode [34] (ftp://ftp.ebi.ac.uk/pub/databases/gencode/Gencode_human/release_28/gencode.v28.transcripts.fa.gz, accessed on 16 May 2018) to establish a mRNA database. The sequencing data were extracted from the probe information site of Agilent Inc. The candidate RNAs, with more than a two-fold change in levels between parental cells and resistant sublines, were detected. The protein linked to the candidate RNAs were detected using RNA-protein binding data (http://starbase.sysu.edu.cn/starbase2/index.php, accessed on 16 May 2018) [35].

## Figures and Tables

**Figure 1 ijms-22-04005-f001:**
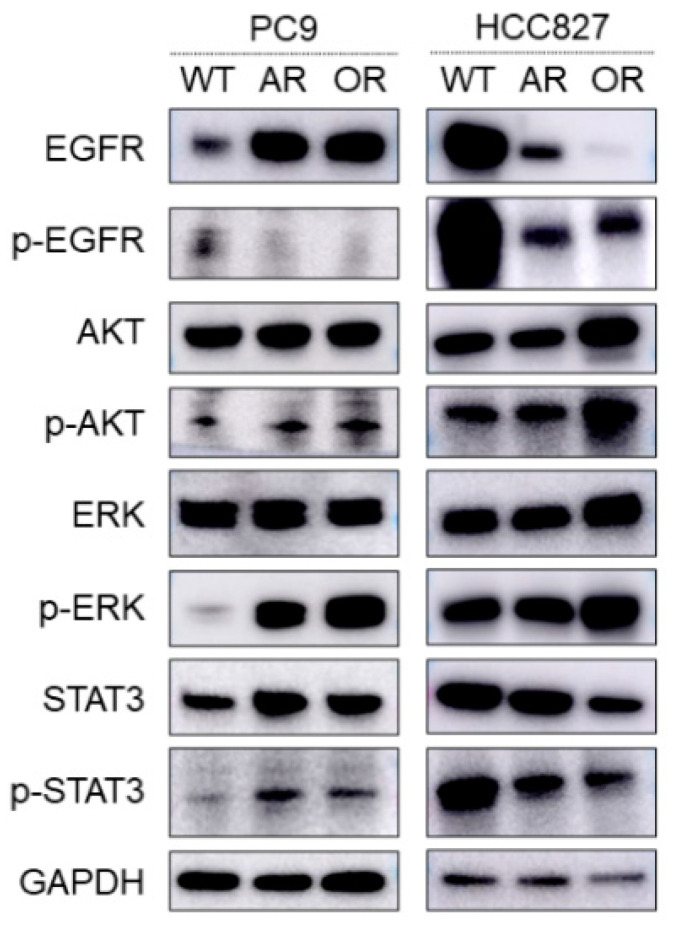
Protein expression levels of EGFR signaling pathway molecules, namely AKT, ERK, and STAT3 in parental cell lines and established EGFR-TKIs-resistant cell lines by Western blotting. PC-9-afatinib-resistant cell line (PC-9-AR); PC-9-osimertinib-resistant cell line (PC-9-OR); HCC827-afatinib-resistant cell line (HCC827-AR); and HCC827-osimertinib-resistant cell line (HCC827-OR).

**Figure 2 ijms-22-04005-f002:**
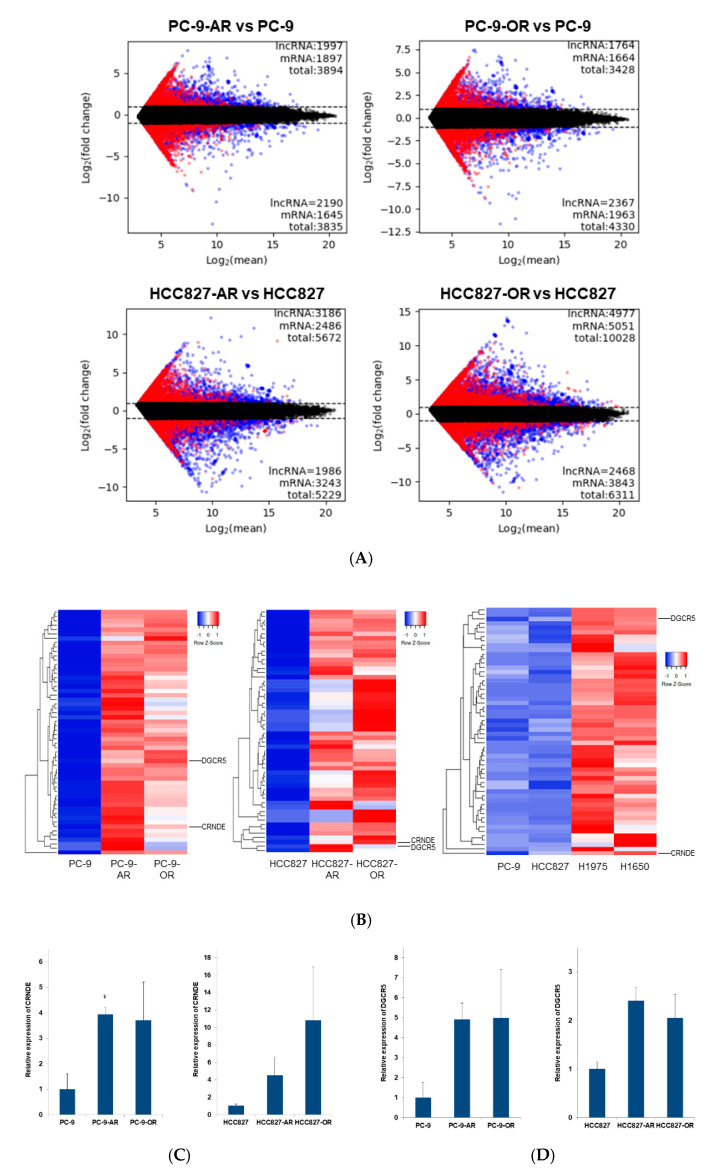
Overexpression of CRNDE and DGCR5 in EGFR-TKIs-resistant NSCLC cells. (**A**) Microarray analyses: The candidate lncRNAs (blue) and mRNAs (red) were determined by differences in expression using a cut-off value of more than two-fold change between resistant and parental cells. (**B**) The expression of CRNDE and DGCR5 lncRNAs was higher in all examined resistant cell lines than in the parental cell lines. In addition, these lncRNAs showed higher expression levels in intrinsic afatinib or osimertinib-resistant cells (i.e., H1975 and H1650) than in sensitive cells (i.e., PC9 and HCC827). (**C**) The expression of CRNDE was higher in all examined resistant cell lines than in the parental cell lines. * *p* < 0.01. (**D**) The expression of DGCR5 was higher in all examined resistant cell lines than in the parental cell lines.

**Figure 3 ijms-22-04005-f003:**
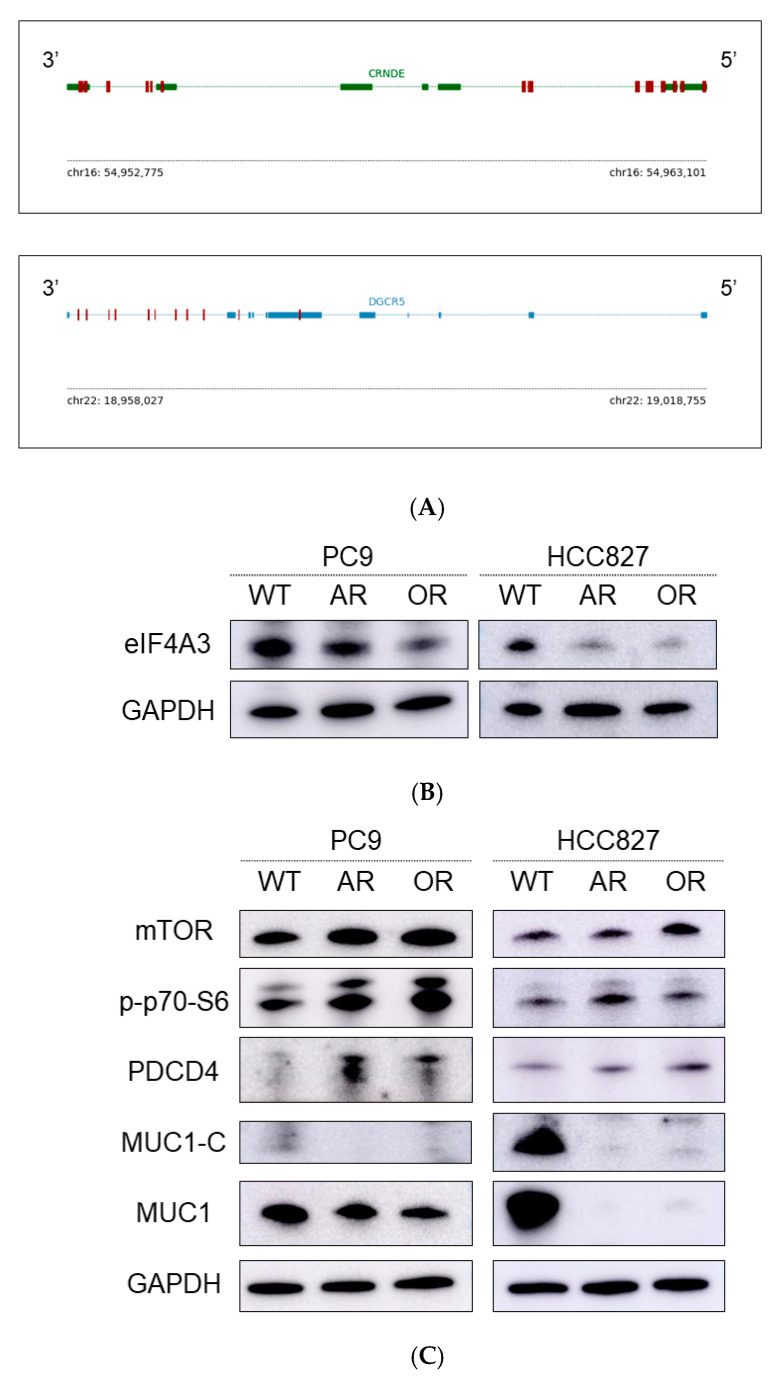
CRNDE decreased the levels of eIF4A3 downstream molecules by binding eIF4A3 in EGFR-TKI-resistant NSCLC cells (**A**) A motif of the eIF4A3-binding site on CRNDE. eIF4A3-binding sites are shown as red rectangles. (**B**) Western blotting analysis showed that eIF4A3 expression declined in the EGFR-TKI-resistant cells. (**C**) Western blotting analysis demonstrated increased protein expression of the AKT/mTOR signaling molecules and decreased protein expression of the eIF4A downstream signaling molecules in EGFR-TKI-resistant cells.

**Figure 4 ijms-22-04005-f004:**
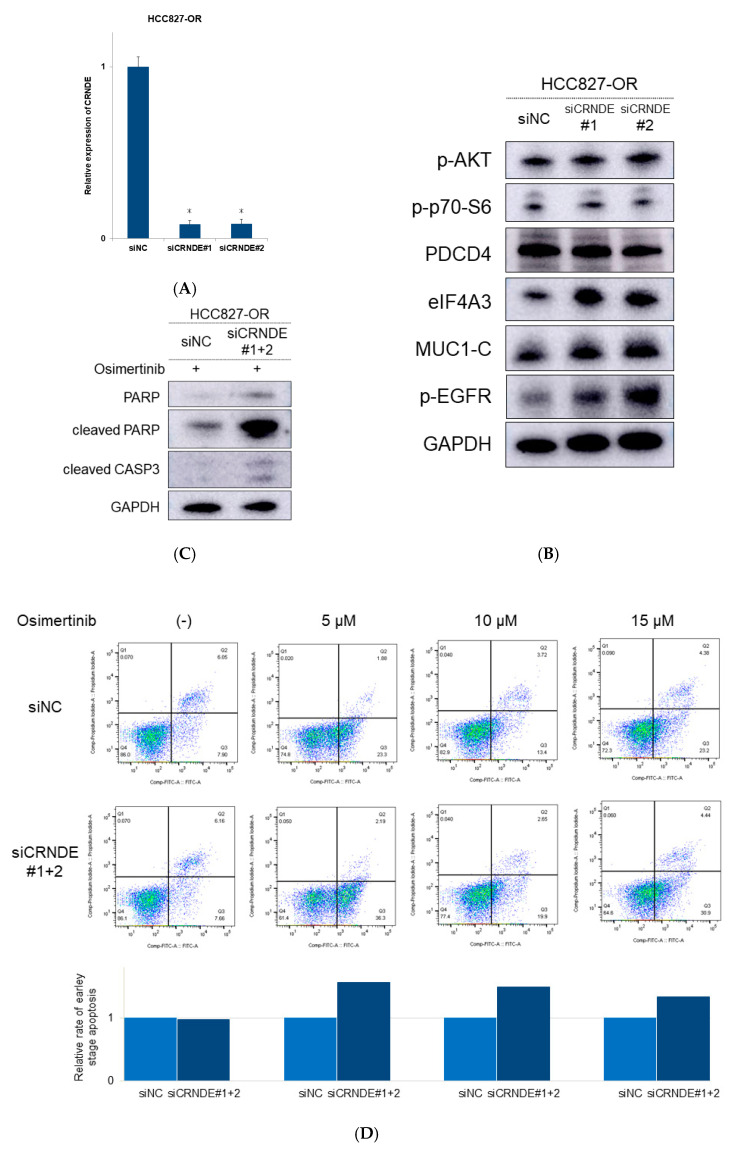
Knockdown of CRNDE induced apoptosis and recovered sensitivity to EGFR-TKIs in drug-resistant cells. (**A**) qRT-PCR showed that transfection of HCC827-OR cells with siRNA because CRNDE inhibited its expression. Fold change for the decrease in CRNDE relative expression. * *p* < 0.01. (**B**) Western blotting analysis showed constant protein expression of the AKT/mTOR signaling molecules and increased protein expression of the eIF4A3 downstream signaling molecules after knockdown of CRNDE in HCC827-OR cells. (**C**) Western blotting analysis showed overexpression of apoptosis-related proteins after treatment with osimertinib following knockdown of CRNDE in HCC827-OR cells. (**D**) Apoptosis induced by osimertinib at some concentrations after knockdown of CRNDE in HCC827-OR cells, as determined by annexin V-propidium iodide (PI) staining. (**E**) Dose-dependent sensitivity of EGFR-TKI-resistant cell lines to afatinib and osimertinib after knockdown of CRNDE by siRNA.

**Figure 5 ijms-22-04005-f005:**
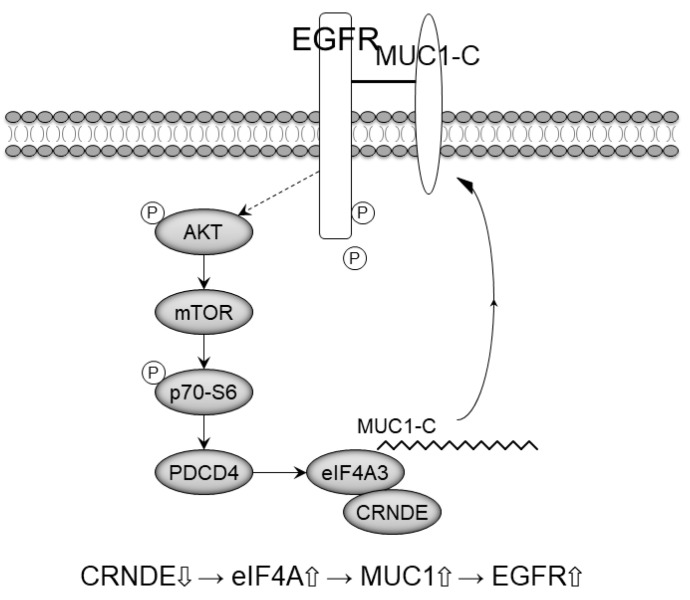
Schematic results. In EGFR-TKI-resistant cells, the expression levels of eIF4A3 and its downstream signaling molecules (i.e., MUC1 and p-EGFR) were decreased, whereas those of its upstream molecules (i.e., AKT, mTOR, p70-S6, and PDCD4) were increased. Following inhibition of CRNDE, the expression levels of upstream molecules AKT, mTOR, and p70-S6 were not changed. In contrast, eIF4A/MUC1/p-EGFR expression was increased. The solid arrow indicated a function on the downstream molecule and the dotted arrow showed the loss of its function.

**Table 1 ijms-22-04005-t001:** Characteristics and IC_50_ of parental and established EGFR-TKI-resistant NSCLC cells.

	Afatinib		Osimertinib		Mutation
	IC_50_ Value (μM)	*p*-Value	IC_50_ Value (μM)	*p*-Value	
	(Mean ± SD)		(Mean ± SD)		
PC-9	<0.001		0.004 ± 0.0008		A deletion in exon 19
PC-9-AR	2.3 ± 0.32	<0.001	-		
PC-9-OR	-		2.4 ± 0.10	<0.001	
HCC827	<0.001		<0.001		A deletion in exon 19
HCC827-AR	2.4 ± 0.84	0.016	-		
HCC827-OR	-		3.1 ± 0.58	0.002	
H1975	3.3 ± 1.9		0.136 ± 0.043		L858R/T790M
H1650	2.3 ± 0.33		2.1 ± 0.27		A deletion in exon 19
					/PTEN null

Abbreviations: AR, afatinib-resistant; EGFR-TKI, epidermal growth factor receptor-tyrosine kinase inhibitor; IC_50_, concentration of drug needed for 50% inhibition of growth; OR, osimertinib-resistant; NSCLC, non-small cell lung cancer; SD, standard deviation.

**Table 2 ijms-22-04005-t002:** RNA-protein binding data.

CRNDE			DGCR5
EIF4A3	IGF2BP1	EWSR1	EIF4A3
FMRP	IGF2BP2	FUS-mutant	FMRP
FUS	IGF2BP3	TAF15	FUS
LIN28A	TNRC6	U2AF65	LIN28A
UPF1	FXR2	TIA1	UPF1
DGCR8	LIN28B	TIAL1	SFRS1
HuR	LIN28	hnRNPC	
PTB	ZC3H7B		

## Data Availability

All data generated or analyzed during this study are included in this published article and its Appendix A. Microarray data have been deposited in the NCBI Gene Expression Omnibus (GEO; http://www.ncbi.nlm.nih.gov/geo/, accessed on 29 December 2020) and are accessible through the GEO series accession number GSE163913.

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
