# Peer review of "Long Non-Coding RNA CRNDE Is Involved in Resistance to EGFR Tyrosine Kinase Inhibitor in EGFR-Mutant Lung Cancer via eIF4A3/MUC1/EGFR Signaling"

_ijms, 2021, doi:10.3390/ijms22084005_

Round 1

Reviewer 1 Report

The manuscript by Takahashi et al describes lncRNA CRNDE as a potential player in resistance to EGFR TKI therapy. The authors generated osimertinib and afatinib resistant NSCLC cells and conducted a microarray analysis to identify post-transcriptional regulatory mechanisms asociated with TKI resistance.

Comments: 

  • P values above 0.05 should not be considered as significant differences throughout the manuscript.
  • Results paragraph 1 should not repeat the content of Table 1, it is redundant.
  • Labeling of the sections in Fig 3 (B,C,D) is inconsistent with the legend. 
  • As the authors mention, expression analysis of CRNDE in post-progression clinical samples could greatly increase the value of the paper. If available in the meantime, could be added to the paper.
  • The microarray analysis could be described in more detail in the method section.  
  • Lots of abbreviations at Table 2, could possibly be reduced.
  • The term 'medical-industrial collaboration' is mentioned many more times than actually needed.
  • Typos in the text, someone with fresh eyes should have a look and correct.

Reviewer 2 Report

In this in vitro study, the authors established afatinib or osimertinib acquired resistant cell lines from PC9 and HCC827 cells. The authors did not detect any EGFR secondary mutation or MET amplification in these resistant cells. Through a comprehensive approach, the authors found that increase expression of long non-coding RNA, CRNDE, was the mechanism of acquired resistance. Several major and minor comments from the reviewer are summarized below.

1. Please describe the methods that were used to detect EGFR secondary mutations and MET gene amplification.

2. Several studies reported that H1975 (L858R/T790M) is sensitive to osimertinib. The reviewer thinks that, theoretically, H1975 should be sensitive to osimertinib. Therefore, the reviewer requires the authors to evaluate the efficacy of osimertinib in H1975 cells again (or to check the cell identity of H1975 cells by a DNA finger-print check), and if osimertinib is effective against H1975, the reviewer suggests to revise the sentence in the Results "These lncRNAs were more highly expressed in intrinsically resistant cells (e.g., H1975 and H1650 cells) than in sensitive cells (page 5, line 121)".

3. It is not clear if the lysates of western blotting were obtained in the presence or absence of drug. For example, was the lysate obtained under the influence of osimertinib in HCC827-OR cells in the Figure 1B? Or the lysate was obtained after several days of drug holiday?

4. The size of the total-EGFR bands are different between PC9 parental cells and PC9-AR / PC9-OR cells. Why...?

5. In the Table 2, the authors summarized the name of proteins as targets for the lncRNAs, CRNDE and DGCR5. Could you add additional information about the roles of lncRNA (for example, downregulate the protein, etc).

6. The authors described that "These lncRNAs were presumed to bind directly to eIF4A3, which was the most abundant protein in EGFR-TKI133 resistant cells (page 5)". However, the reviewer cannot agree with this based on the Figure 3B. elF4A3 does not seem to be abundant in the resistant cells.

7. This is the most critical point of this manuscript. The authors described that "Knockdown of CRNDE restored sensitivity to EGFR-TKIs in HCC827-OR and HCC827-AR (page 9)". However, the difference of growth inhibitory curves between CRNDE knock-down cells and control cells was small (Figure 4E), therefore, it is not adequate to describe that knockdown of CRNDE "restored sensitivity" to EGFR-TKI. 

8. In addition, it is strange that cell growth was accelerated in control cells in the presence of lower concentration of osimertinib (Figure 4E, left), that is quite different from the data in the Figure 1A.

9. In Figure 5, the authors provided a schema, however, the reviewer wonders how MUC1C affects EGFR. Does MUC1C bind EGFR? Does MUC1C activate EGFR?
